# Segment Anything Model (SAM) for Digital Pathology: Assess Zero-shot Segmentation on Whole Slide Imaging

**Ruining Deng**[*][1]                                                     R.DENG@VANDERBILT.EDU

[1] *Vanderbilt University, Nashville, TN, USA*

**Can Cui**[*][1]                                                         CAN.CUI.1@VANDERBILT.EDU

**Quan Liu**[*][1]                                                       QUAN.LIU@VANDERBILT.EDU

**Tianyuan Yao**[1]                                                     TIANYUAN.YAO@VANDERBILT.EDU

**Lucas W. Remedios**[1]                                             LUCAS.W.REMEDIOS@VANDERBILT.EDU

**Shunxing Bao**[1]                                                   SHUNXING.BAO@VANDERBILT.EDU

**Bennett A. Landman**[1]                                           BENNETT.LANDMAN@VANDERBILT.EDU

**Lee E. Wheless**[2,3]                                               LEE.E.WHELESS@VUMC.ORG

[2] *Vanderbilt University Medical Center, Nashville, TN, USA*

[3] *Veterans Affairs Tennessee Valley Healthcare System, Nashville, TN, USA*

LEE.E.WHELESS@VUMC.ORG

**Lori A. Coburn**[2,3]                                               LORI.COBURN@VUMC.ORG

**Keith T. Wilson**[2,3]                                             KEITH.WILSON@VUMC.ORG

**Yaohong Wang**[2]                                                   YAOHONG.WANG@VUMC.ORG

**Shilin Zhao**[2]                                                     SHILIN.ZHAO.1@VUMC.ORG

**Agnes B. Fogo**[2]                                                   AGNES.FOGO@VUMC.ORG

**Haichun Yang**[2]                                                   HAICHUN.YANG@VUMC.ORG

**Yucheng Tang**[4]                                                   YUCHENGT@NVIDIA.COM

[4] *NVIDIA Cooperation, Redmond, WA, USA*

**Yuankai Huo**[†][1]                                                 YUANKAI.HUO@VANDERBILT.EDU

**Editors:** Under Review for MIDL 2023

## Abstract

The segment anything model (SAM) was released as a foundation model for image segmentation. The promptable segmentation model was trained by over 1 billion masks on 11M licensed and privacy-respecting images. The model supports zero-shot image segmentation with various segmentation prompts (e.g., points, boxes, masks). It makes the SAM attractive for medical image analysis, especially for digital pathology where the training data are rare. In this study, we evaluate the zero-shot segmentation performance of SAM model on representative segmentation tasks on whole slide imaging (WSI), including (1) tumor segmentation, (2) non-tumor tissue segmentation, (3) cell nuclei segmentation. **Core Results:** *The results suggest that the zero-shot SAM model achieves remarkable segmentation performance for large connected objects. However, it does not consistently achieve satisfying performance for dense instance object segmentation, even with 20 prompts (clicks/boxes) on each image.* We also summarized the identified limitations for digital pathology: (1) image resolution, (2) multiple scales, (3) prompt selection, and (4) model fine-tuning. In the future, the few-shot fine-tuning with images from downstream pathological segmentation tasks might help the model to achieve better performance in dense object segmentation.

**Keywords:** segment anything, SAM model, digital pathology, medical image analysis.

---

[*] Joint first author: contributed equally

[†] Corresponding author

## 1. Introduction

Large language models (e.g., ChatGPT (Brown et al., 2020) and GPT-4 (OpenAI, 2023)), are leading a paradigm shift in natural language processing with strong zero-shot and few-shot generalization capabilities. Segmenting objects (e.g., tumor, tissue, cell nuclei) for whole slide imaging (WSI) data is an essential task for digital pathology (Huo et al., 2021). The "Segment Anything Model" (SAM) (Kirillov et al., 2023) was proposed as a foundation model for image segmentation. The model has been trained on over 1 billion masks on 11 million licensed and privacy-respecting images. Furthermore, the model supports zero-shot image segmentation with various segmentation prompts (e.g., points, boxes, and masks). This feature makes it particularly attractive for pathological image analysis where the labeled training data are rare and expensive.

In this study, we assess the zero-shot segmentation performance of the SAM model on representative segmentation tasks, including (1) tumor segmentation (Liu et al., 2021), (2) tissue segmentation (Deng et al., 2023), and (3) cell nuclei segmentation (Li et al., 2021). Our study reveals that the SAM model has some limitations and performance gaps compared to state-of-the-art (SOTA) domain-specific models.

## 2. Experiments and Performance

We obtained the source code and the trained model from https://segment-anything.com. To ensure scalable assessments, all experiments were performed directly using Python, rather than relying on the Demo website. The results are presented in Figure 1 and Table 1.

**Tumor Segmentation**. We employed SimTriplet (Liu et al., 2021) approach as the SOTA method, with the same testing cohort to make a fair comparison. In order to be compatible with the SAM segmentation model, the WSI inputs were scaled down 80 times from a resolution of $40\times$, resulting in an average size of $860\times1279$ pixels. **Tissue Segmentation**. We employed Omni-Seg (Deng et al., 2023) approach as the SOTA method, with the same testing cohort to make a fair comparison.. The tissue types consist of the glomerular unit (CAP), glomerular tuft (TUFT), distal tubular (DT), proximal tubular (PT), arteries (VES), and peritubular capillaries (PTC). **Cell nuclei Segmentation**. The MoNuSeg dataset (Kumar et al., 2019) includes 30 images for training and 14 for testing. We evaluated the performance of SAM models against the BEDs model (Li et al., 2021), a competitive nuclei segmentation model trained on the MoNuSeg training data.

## 3. Limitations on Digital Pathology

The SAM models achieve remarkable performance under zero-shot learning scenarios. However, we identified several limitations during our assessment.

**Image resolution**. The average training image resolution of SAM is $3300\times4950$ pixels (Kirillov et al., 2023), which is significantly smaller than Giga-pixel WSI data ($> 10^9$ pixels). **Multiple scales**. Multi-scale is a significant feature in digital pathology. Different tissue types have their optimal image resolution (as shown in Table 1). **Prompt selection**. To achieve decent segmentation performance in zero-shot learning scenarios, a considerable number of prompts are still necessary. **Model fune-tuning**. A reasonable online/offline

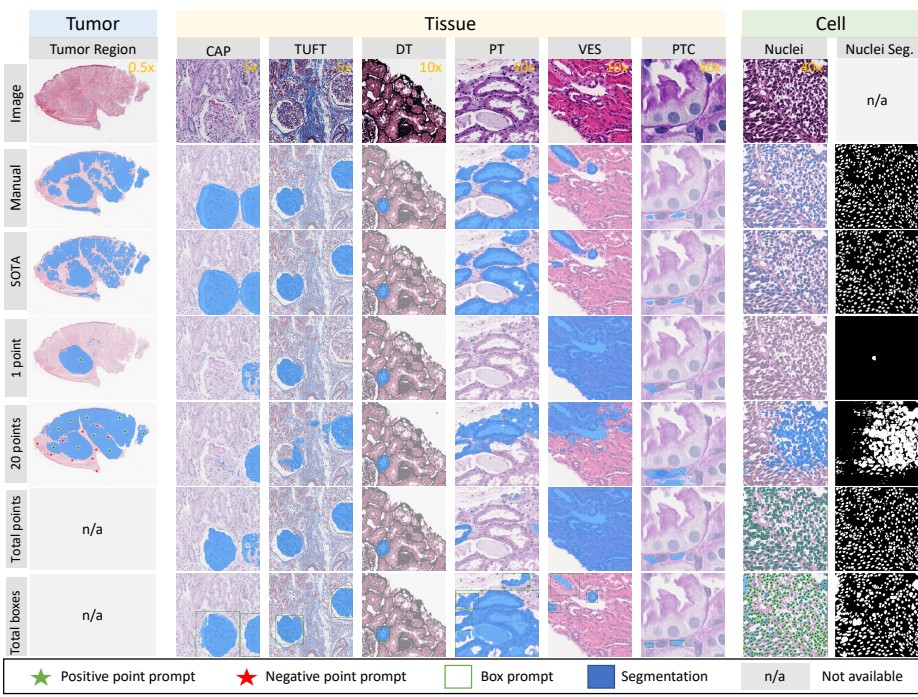

Figure 1: **Qualitative segmentation results**. The SOTA methods are compared with SAM method with different prompt strategies.

Table 1: Compare SAM with state-of-the-art (SOTA) methods. (Unit: Dice score)

| Method | Prompts | Tumor | Tissue | | | | | | | Cell |
|---|---|---|---|---|---|---|---|---|---|---|
| | | 0.5× | 5× | | 10× | | | 40× | 40× | |
| | | Tumor | CAP | TUFT | DT | PT | VES | PTC | Nuclei | |
| SOTA | no prompt | 71.98 | 96.50 | 96.59 | 81.01 | 89.80 | 85.05 | 77.23 | 81.77 | |
| SAM | 1 point | 58.71 | 78.08 | 80.11 | 58.93 | 49.72 | 65.26 | 67.03 | 1.95 | |
| SAM | 20 points | 74.98 | 80.12 | 79.92 | 60.35 | 66.57 | 68.51 | 64.63 | 41.65 | |
| SAM | total points | n/a | 88.10 | 89.65 | 70.21 | 73.19 | 67.04 | 67.61 | 69.50 | |
| SAM | total boxes | n/a | 95.23 | 96.49 | 89.97 | 86.77 | 87.44 | 87.18 | 88.30 | |

total points/boxes: we place points/boxes on every single instance object (based on the known ground truth) as a theoretical upper bound of SAM. Note that it is impractical in real applications.

fine-tuning strategy is necessary to propagate the knowledge obtained from manual prompts to larger-scale automatic segmentation on Giga-pixel WSI data.

**Acknowledgements**.This research was supported by NIH R01DK135597, The Leona M. and Harry B. Helmsley Charitable Trust grant G-1903-03793 and G-2103-05128, NSF CA-

REER 1452485, NSF 2040462, NCRR Grant UL1 RR024975-01 (NCATS Grant 2 UL1 TR000445-06), NIH NIDDK DK56942, DoD HT94252310003, the VA grants I01BX004366 and I01CX002171, VUMC Digestive Disease Research Center supported by NIH grant P30DK058404, NVIDIA hardware grant, resources of ACCRE at Vanderbilt University.

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
