# OpenReview forum: "Segment Anything Model (SAM) for Digital Pathology: Assess Zero-shot Segmentation on Whole Slide Imaging"
_MIDL.io/2023/Short_Paper_Track — MIDL 2023 Short paper track Poster_

### Official Review · Reviewer_WpnM · 2023-04-10
**The study evaluates the zero-shot segmentation performance of the Segment Anything Model (SAM) on representative segmentation tasks in digital pathology, including tumor segmentation, tissue segmentation, and cell nuclei segmentation.**

**Rating:** 6
**Confidence:** 3

**Review:**

The study evaluates the zero-shot segmentation performance of the Segment Anything Model (SAM) on representative segmentation tasks in digital pathology, including tumor segmentation, tissue segmentation, and cell nuclei segmentation. The SAM model achieves remarkable performance for large connected objects but has limitations and performance gaps for dense instance object segmentation. The study identifies limitations such as image resolution, multiple scales, prompt selection, and model fine-tuning. The SAM model's zero-shot learning capability and support for various segmentation prompts make it attractive for medical image analysis, especially in digital pathology where labeled training data are rare.

Pros:
The SAM model has been trained on over 1 billion masks on 11 million licensed and privacy-respecting images, making it a strong foundation model for image segmentation.
The model supports zero-shot image segmentation with various segmentation prompts, making it attractive for medical image analysis, especially for digital pathology where labeled training data are rare.
The SAM model achieves remarkable segmentation performance for large connected objects.
The study provides a fair comparison with state-of-the-art (SOTA) domain-specific models for representative segmentation tasks on whole slide imaging (WSI).

Cons:
The technical development is limited.
The SAM model does not consistently achieve satisfying performance for dense instance object segmentation, even with multiple prompts on each image.
The model has limitations and performance gaps compared to SOTA domain-specific models.
The average training image resolution of SAM is significantly smaller than giga-pixel WSI data, which is commonly encountered in digital pathology.
Multi-scale is a significant feature in digital pathology, and the model may not perform optimally for different tissue types.
A considerable number of prompts are still necessary to achieve decent segmentation performance in zero-shot learning scenarios.
A reasonable online/offline fine-tuning strategy is necessary to propagate the knowledge obtained from manual prompts to larger-scale automatic segmentation on giga-pixel WSI data.

---

### Official Review · Reviewer_M2Ni · 2023-04-24
**Needed and useful benchmark**

**Rating:** 8
**Confidence:** 4

**Review:**

This paper uses the Times New Roman font (package times), and as it is a more compact font, it technically over the page limit: it could be a desk reject. At the same time, the template is very space-hungry for long author lists, so I suppose that balance things out, not everyone has the possibility to have a short author list.

As for the actual review:

This paper performs a benchmark of the Segment Anything Model, on three different tasks (tumor, tissue, and cells segmentation), evaluating how it performs (with various prompts) compared to the respective state of the arts. The results (in my view) show that SAM has a very high potential as an interactive annotation tool, much less as a end-to-end method.